# Alteration of *Salmonella enterica* Virulence and Host Pathogenesis through Targeting *sdiA* by Using the CRISPR-Cas9 System

**DOI:** 10.3390/microorganisms9122564

**Published:** 2021-12-11

**Authors:** Momen Askoura, Ahmad J. Almalki, Amr S. Abu Lila, Khaled Almansour, Farhan Alshammari, El-Sayed Khafagy, Tarek S. Ibrahim, Wael A. H. Hegazy

**Affiliations:** 1Department of Microbiology and Immunology, Faculty of Pharmacy, Zagazig University, Zagazig 44519, Egypt; 2Department of Pharmaceutical Chemistry, Faculty of Pharmacy, King Abdulaziz University, Jeddah 21589, Saudi Arabia; ajalmalki@kau.edu.sa (A.J.A.); tmabrahem@kau.edu.sa (T.S.I.); 3Center of Excellence for Drug Research and Pharmaceutical Industries, King Abdulaziz University, Jeddah 21589, Saudi Arabia; 4Department of Pharmaceutics and Industrial Pharmacy, Faculty of Pharmacy, Zagazig University, Zagazig 44519, Egypt; a.abulila@uoh.edu.sa; 5Department of Pharmaceutics, College of Pharmacy, University of Hail, Hail 81442, Saudi Arabia; kh.almansour@uoh.edu.sa (K.A.); frh.alshammari@uoh.edu.sa (F.A.); 6Department of Pharmaceutics, College of Pharmacy, Prince Sattam Bin Abdulaziz University, Al-kharj 11942, Saudi Arabia; e.khafagy@psau.edu.sa; 7Department of Pharmaceutics and Industrial Pharmacy, Faculty of Pharmacy, Suez Canal University, Ismailia 41552, Egypt

**Keywords:** *Salmonella enterica*, CRISPR-Cas9, *sdiA*, *ssaV*, virulence, pathogenesis

## Abstract

*Salmonella enterica* is a common cause of many enteric infections worldwide and is successfully engineered to deliver heterologous antigens to be used as vaccines. Clustered Regularly Interspaced Short Palindromic Repeats (CRISPRs) RNA-guided Cas9 endonuclease is a promising genome editing tool. In the current study, a CRISPR-Cas9 system was used to target *S.*
*enterica* *sdiA* that encodes signal molecule receptor SdiA and responds to the quorum sensing (QS) signaling compounds N-acylhomoserine lactones (AHLs). For this purpose, *sdiA* was targeted in both *S.*
*enterica* wild type (WT) and the Δ*ssaV* mutant strain, where SsaV has been reported to be an essential component of SPI2-T3SS. The impact of *sdiA* mutation on *S.* *enterica* virulence was evaluated at both early invasion and later intracellular replication in both the presence and absence of AHL. Additionally, the influence of *sdiA* mutation on the pathogenesis *S.* *enterica* WT and mutants was investigated in vivo, using mice infection model. Finally, the minimum inhibitory concentrations (MICs) of various antibiotics against *S.* *enterica* strains were determined. Present findings show that mutation in *sdiA* significantly affects *S.*
*enterica* biofilm formation, cell adhesion and invasion. However, *sdiA* mutation did not affect bacterial intracellular survival. Moreover, in vivo bacterial pathogenesis was markedly lowered in *S.*
*enterica* Δ*sdiA* in comparison with the wild-type strain. Significantly, double-mutant *sdiA* and *ssaV* attenuated the *S. enterica* virulence and in vivo pathogenesis. Moreover, mutations in selected genes increased *Salmonella* susceptibility to tested antibiotics, as revealed by determining the MICs and MBICs of these antibiotics. Altogether, current results clearly highlight the importance of the CRISPR-Cas9 system as a bacterial genome editing tool and the valuable role of SdiA in *S.*
*enterica* virulence. The present findings extend the understanding of virulence regulation and host pathogenesis of *Salmonella*
*enterica*.

## 1. Introduction

*Salmonella enterica* are facultative anaerobic intracellular Gram-negative non-lactose fermenting motile bacteria that belong to the family Enterobacteriaceae. *S. enterica* infections greatly vary from a mild gastroenteritis, caused mostly by *S. enterica* serovars Typhimurium (*S.* Typhimurium) and Enteritidis (*S.* Enteritidis), to serious systemic infections of typhoid fever caused by *S. enterica* serovar Typhi (*S.* Typhi) or Paratyphi (*S.* Paratyphi) [1]. The encoding genes for numerous significant virulence factors of *S. enterica* are arranged in specific loci called *Salmonella* Pathogenicity Islands (SPI) [2]. *Salmonella* deploys intricate virulence factors named type III secretion systems (TTSS), which mediate distinct functions [3]. Two major SPI encode TTSS to translocate *Salmonella* effectors in different phases of pathogenesis [4]. SPI1-TTSS translocates TTSS effector proteins into host cell cytoplasm during early stages of invasion, while SPI2-TTSS translocates TTSS effector proteins responsible for bacterial intracellular survival at later stages [2,5]. Previous studies described the role of the SPI2-T3SS machinery component SsaV and its importance for the secretion of most T3SS effectors [6,7,8]. Importantly, mutations in *ssaV* could lead to a significant reduction in *S. enterica* virulence through decreasing the translocation of SPI2-effector proteins, as this decrease affects the ability of *S. enterica* to survive intracellularly [7].

Quorum sensing (QS) is a way that bacteria use autoinducer (AI) molecules, such as N-acylhomoserine lactones (AHLs), for cell-to-cell communication, which plays a crucial role in bacterial virulence [9,10,11]. It has been shown that *S. enterica* contains at least two types of QS systems, one is induced by acylhomoserine lactone (AHL) and the other is induced by autoinducer-2 (AI-2) [12]. *S. enterica* employs QS to enhance bacterial virulence and pathogenesis through regulation of biofilm formation, virulence factors’ production and swarming motility [13,14]. *S. enterica* does not encode an AHL synthase, but it encodes SdiA, a LuxR homolog, which detects AHLs. A variety of AHL molecules with different acyl chain length and substituents at the C-3 position have been reported to mediate QS [15]. SdiA detects solely AHLs produced by other bacterial species and therefore plays a significant role in QS [16,17].

Recently, *S. enterica* was used as a carrier to deliver heterologous antigen fusions to stimulate both humoral and cellular immune responses [1,18]. *S. enterica* was engineered to be a candidate for bacteria-mediated tumor therapy [19,20]. The approved safety of *S. enterica* mutants, as well as other factors, makes these bacteria a promising carrier for vaccination against both bacterial, viral infections and cancer, as well [1]. Editing of *S. enterica* chromosome is essential in order to develop new mutant strains which can be used efficiently as carriers for vaccination purposes or used themselves as vaccines [1,19]. Interestingly, affordable and efficient genome editing tools have been developed recently in order to engineer both eukaryotic and prokaryotic organisms. 

The CRISPR RNA guided endonuclease is a promising and efficient genome editing tool [21,22]. The CRISPR-Cas system was discovered as a naturally occurring adaptive microbial immune system against invading viruses and other mobile genetic elements [23,24]. Importantly, the CRISPR-Cas9 system was successfully used in targeting the genome of both bacterial [25,26,27,28] and eukaryotic cells [29,30,31]. Mutagenesis introduces a selection marker in the edited locus or requires a process of two steps that includes a counter system for selection [32]. Genome editing tools, such as zinc finger nucleases (ZFN), transcription activator-like effector nucleases (TALENs) and homing meganucleases, have been programmed to cut genomes in specific locations. However, these engineering techniques have been reported to be difficult to use and expensive [25,33].

The CRISPR loci consist of a repeated array of short sequences separated by short spacer sequences; these spacer sequences are complementary to genomes of invading viruses, as well as bacterial and archaeal plasmids [34,35,36,37]. The CRISPR-Cas immunity system occurs in three stages: First, Cas proteins integrate short sequences of invading DNA into CRISPR array as a new spacers [38]. Second, as a consequence, the CRISPR array will be transcribed and processed to produce small CRISPR RNAs (crRNAs) that contain a spacer sequence. Finally, crRNAs in association with Cas nucleases target the spacer sequence, leading to its cleavage resulting in destruction of invader’s DNA [23,24,39]. There are three major types of prokaryotic CRISPR immune that are grouped according to operon organization and *cas* gene conservation [39]: The type II CRISPR-Cas system is characterized by RNA-guided Cas9 endonuclease activity. It is the simplest of all Cas systems to be used to interfere or even edit both eukaryotic or prokaryotic genomes [31,40,41]. The Cas9 endonuclease activity requires guide sequence (crRNA) to guarantee precise targeting, as well as an immediate downstream motif sequence (PAM). In order to edit the bacterial genome, it is necessary to transfer a vector encoding Cas9 and its guide and recombination template containing the desired mutation [25]. The spacer or PAM sequences must be altered in order to prevent re-cleavage of Cas9 of target genome. This approach has been efficiently used to manipulate several bacterial species [25,26,42].

The current study investigated the effect of *sdiA* mutation on *S. enterica* pathogenesis. The virulence of both *S. enterica* wild type (WT) and Δ*ssaV* mutant is evaluated herein. The *S. enterica* Δ*ssaV* mutant has been studied as a carrier for vaccination [32,43,44]. In addition, *S. enterica* chromosome was edited by using an efficient CRISPR-Cas9 system. Moreover, *sdiA,* which plays an essential role in QS, was targeted in two sites, using Cas9 encoding plasmids in both *S. enterica* WT and *ssaV* mutants. This study aimed to elucidate how much the mutation in *sdiA* and *ssaV* separately, as well as double mutation, would affect *Salmonella* virulence. The influence of *sdiA* mutation on the pathogenesis of both *S. enterica* WT and Δ*ssaV* mutants in early stages of invasion and intracellular survival are characterized. Finally, the effect of *sdiA* mutation on biofilm formation, susceptibility to antibiotics and in vivo pathogenesis are characterized. 

## 2. Materials and Methods

### 2.1. Bacterial Strains, Plasmids Enzymes, Media and Chemicals

*S. enterica* serovar Typhimurium NCTC 12023, and the *S.* Typhimurium Δ*ssaV* mutant were kindly provided by Hensel’s lab (Germany). Plasmids pCRISPR and pCas9 were obtained from Addgene (http://www.addgene.org/, accessed on 12 May 2021) with No. 42875 and 42876, respectively [25]. Plasmids were introduced into *S. enterica* strains by electroporation, and recombinant strains were cultured in medium containing kanamycin (50 µg/mL), or chloramphenicol (25 µg/mL). All enzymes used to clone CRISPR plasmids and restriction endonuclease were provided from New England Biolabs, USA. Tryptone soy broth (TSB), Tryptic Soy Agar (TSA), Mueller Hinton (MH) broth and agar and Luria–Bertani (LB) broth and agar were purchased from Oxoid (Hampshire, UK). Dulbecco′s Modified Eagle′s Medium (DMEM) medium was obtained from Sigma-Aldrich (St. Louis, MO, USA). The used N-acylhomoserine lactones is N-hexanoyl-DL-homoserine lactone (CAS Number: 106983-28-2) was ordered from Sigma-Aldrich (St. Louis, MO, USA). All used chemicals were of pharmaceutical grade.

### 2.2. Targeting sdiA by CRISPR/Cas9

Two plasmids were employed: the first plasmid, pCas9, encodes the Cas9, trcrRNA and crRNA to target guide sequence number 1. The other plasmid, pCRISPR, encodes crRNA for guide sequence number 2 to be targeted by Cas9. It was shown that mutation induction can be facilitated by the co-selection of transformable cells and use of dual-RNA:Cas9 cleavage to induce a small induction of recombination at the target locus. Both plasmids pCas9 and pCRISPR were transformed to competent cells, followed by selection on kanamycin and chloramphenicol-containing LB [25]. 

The guide sequences shown in Figure 1 were chosen for targeting *sdiA*, using a CRISPER/Cas system. Plasmids pCas9 and pCRISPR were digested by *Bsa*I restriction endonuclease, and digested plasmids were gel-purified. The protocol provided by Addgene was followed to clone a spacer sequence into pCas9 and pCRISPR. Briefly, a spacer sequence of 20 bp was chosen upstream to NGG to be targeted by Cas9 nuclease and was designed with *Bsa*I restriction cut site ends to be ligated directly to *Bsa*I-digested pCas9 and pCRISPR. Oligonucleotides used for plasmids construction are listed in (Table 1). Oligo nucleotides I and II were designed to target the first site, and Oligos III and IV were designed to target the second site (Table 1). The Oligo nucleotides ordered to by synthesized from Sigma Custom DNA Oligos (St. Louis, MO, USA). Oligos I, II, III and IV were phosphorylated using T4 PNK enzyme. Phosphorylated oligo I was annealed to oligo II, and oligo III was annealed to oligo IV in 1M NaCl at 95 °C for 5 min and slowly cooled down to room temperature. Diluted annealed oligos I and II were ligated to *Bsa*I-digested pCas9 plasmid, and the diluted annealed oligos III and IV were ligated to *Bsa*I-digested pCRISPR plasmid. The ligated plasmids to spacer sequences were electroporated sequentially to *S. enterica* WT and ∆*ssaV* mutant competent cells. The transformed cells were grown at 37 °C for 1 h in LB broth containing kanamycin (50 µg/mL) and chloramphenicol (25 µg/mL). Then 100 µL was spread over LB agar containing kanamycin (50 µg/mL) and chloramphenicol (25 µg/mL) and incubated overnight at 37 °C to select the proper clones that harbor the plasmids carrying resistant genes to these antibiotics. For confirmation of proper cloning, the negative colony PCR clones, using oligo I or oligo III, and sdiA-Rev primer were chosen. PCR products were visualized by electrophoresis on agarose gel (0.7%), using 1X TAE (Tris-acetate-EDTA) running buffer at 80–120V, and visualized by 0.5 g/mL ethidium bromide. 

### 2.3. Adhesion Assay

Overnight cultures of *S. enterica* serovar Typhimurium (*S.* Typhimurium) WT and ∆*ssaV* mutant with or without *sdiA* targeted CRISPR-Cas9 (Δ*sdiA*) strains were prepared, diluted with fresh TSB and adjusted to a cell density of 1 × 10^6^ CFU/mL (OD_600_ = 0.4) for adhesion assay, as previously described [45].

#### 2.3.1. Adhesion to Epithelial Cells

Monolayers of HeLa cells were cultured in 24-well plates in DMEM medium [46,47]. HeLa cells were passaged with 70% confluent and washed with sterile PBS before adhesion assay. Bacterial cultures *S.* Typhimurium WT, ∆*ssaV*, Δ*sdiA* and ∆*ssaV*Δ*sdiA* (1 × 10^6^ CFU/mL) and DMEM with or without N-hexanoyl-DL-homoserine lactone (AHL) in final concentration 0.001 µM were added to wells. Incubation was continued for 1 h at 37 °C. Next, epithelial cells were washed 3 times with PBS and lysed at room temperature for 20 min in Triton X-100 (1%). The bacterial suspensions were serially diluted, plated on TSA and incubated overnight at 37 °C for colony counting. The bacterial counts were used to evaluate adhesion rate. Experiment was performed in triplicate, and the means and standard deviations were calculated.

#### 2.3.2. Adhesion to Abiotic Surface and Biofilm Formation

*S.* Typhimurium strains; WT, ∆*ssaV*, Δ*sdiA* and ∆*ssaV*Δ*sdiA* were cultured with or without N-hexanoyl-DL-homoserine lactone AHL (0.001 µM) in polystyrene microtiter plate and incubated at 37 °C either for 1 h (for evaluation of adhesion) or for 24 h (for evaluation of biofilm formation) [45,47,48,49]. Incubated plates were washed gently 3 times with phosphate buffer saline (PBS), fixed at 60 °C for 25 min, stained with crystal violet (0.1%) for 15 min and finally washed with PBS. The adhered crystal violet was extracted with ethanol, and optical densities were measured at 590 nm. The assay was repeated in triplicate, and results were expressed as the means ± standard deviations.

### 2.4. Invasion Assay and Intracellular Replication 

Internalization of *S.* Typhimurium strains within different cell lines was evaluated by using the gentamicin protection assay, as formerly described [50]. Briefly, 24-wells polystyrene plates were seeded with HeLa cells and/or RAW264.7 at cell density of 5 × 10^5^ and 2 × 10^5^ cells/well, respectively. Tested strains WT, ∆*ssaV*, Δ*sdiA* and ∆*ssaV*Δ*sdiA* were subcultured from overnight cultures and incubated at 37 °C for 4 h to induce SPI1 conditions. A master-mix of the inoculum (1 × 10^5^ bacteria/well) multiplicity of infection (MOI 1) for HeLa cell infection or raw macrophage was prepared in DMEM, and 300 μL was added to each well. The bacterial infections were performed in either in the absence or presence of N-hexanoyl-DL-homoserine lactone AHL (0.001 µM). Non-internalized bacteria were washed out with pre-warmed PBS after 30 min, and the adhered extracellular bacteria were killed by incubation in media containing gentamicin (100 µg/mL) for 1 h. For invasion assays, HeLa cells were lysed with 0.1% Triton X-100 for 10 min at 25 °C. To determine intracellular bacteria, the inoculum and the lysates were serially diluted and plated onto Mueller Hinton (MH) plates. The percentage of invading *Salmonella* (1 h against inoculum × 100) was calculated. To assay the intracellular replication, the infected cells were washed with PBS and lysed with TritonX-100 (0.1%) for 10 min in 25 °C at 2 and 16 h post-infection. The inoculum and the lysates were serially diluted and plated onto MH plates. The phagocytosed cells numbers/relative untaken cells (2 h against inoculum × 100) and x-fold intracellular replication (16 h against 2 h) were evaluated.

### 2.5. The Intracellular Behavior of Salmonella Mutants

#### 2.5.1. Construction of SPI2 Expressing Plasmid

For testing the effectiveness of SPI2-T3SS-dependent translocation, pWSK9 P*_sseJ_sseJ*::hSurvivin::HA plasmid was generated as previously described [18]. The hSurvivin gene was PCR-amplified by employing primers hSurvivin-HA-Rev-*Xba*I and hSurv-For-*Eco*RV, and template plasmid pWSK29 P*_sseA_sscB*sseF::hSurvivin::HA (provided kindly by Prof. Hensel, University of Osanabrueck, Germany). The obtained hSurvivin and pWSK29 plasmid were digested with *Xba*I and *Eco*RV and ligated together. The *sseJ* gene was PCR-amplified by using SseJ-Rev-*Eco*RV and SseJ-Pro-For-*Kpn*I primers prior to its digestion with *Kpn*I and *Eco*RV. The *sseJ* gene and pWSK29::hSurvivin were digested with *Kpn*I and *Eco*RV and ligated to obtain plasmids pWSK29 P*_sseJ_sseJ*::hSurvivin::HA. Constructed plasmid was electroporated in *S.* Typhimurium strains WT, ∆*ssaV*, Δ*sdiA* and ∆*ssaV*Δ*sdiA* component cells. Positive clones were selected on LB containing carbenicillin (50 µg/mL). Obtained plasmid was confirmed by colony PCR and diagnostic digestion, and they were sequenced by using T7-Seq and T3-Seq primers [18]. 

#### 2.5.2. Evaluation of SPI2 Effectors Expression

Plasmid pWsk29 P*_sseJ_sseJ*::hSurvivin was transferred to WT, ∆*ssaV*, Δ*sdiA* and ∆s*saV*Δ*sdiA* strains. Tested mutants and expression rates were analyzed as described before [18]. Briefly, tested strains harboring plasmid expressing SPI2 effector protein SseJ-hSurvivin tagged with HA regulated by P*sseJ* promoter were cultured in SPI2-inducing minimal media (PCN-P, pH 5.8). Bacterial cells were collected by centrifugation after 6 h. Equal amounts of bacterial cells were lysed and exposed to SDS-PAGE. Western blots were used to detect HA epitope tag, using fluorescent-labeled secondary antibodies. The signal intensities were measured by using the Odyssey system (Li-Cor) in comparison to control DnaK (cytosolic heat shock protein). The experiment was performed in triplicate, and ratios of HA/DnaK signals were calculated and expressed as means ± standard deviation. 

#### 2.5.3. Evaluation of Translocation Efficiency 

*S. enterica* WT, ∆*ssaV*, Δ*sdiA* and ∆*ssaV*Δs*diA* provided with constructed plasmid for the expression of HA tagged SPI2 effector were used to infect raw macrophage or HeLa cells in absence or presence of AHL at MOI of 100, as described previously [18]. Briefly, cells were fixed at 16 h after infection; *Salmonella* LPS (rabbit anti-*Salmonella* O1,4,5, Difco, BD) and the HA epitope tag (Roche, Basel, Switzerland) were immuno-stained. The cells were analyzed by microscopy, using a Leica laser-scanning confocal microscope. The fluorescence intensities of tagged protein were detected by J-image program in HeLa cells and macrophages. Infected cells harboring similar number of intracellular bacteria were chosen, and the signal of fluorescence intensities for HA tagged proteins were measured. The mean signal intensities and standard deviations were calculated for at least 30 infected cells per tested strains.

### 2.6. The Effect on Mutation on Bacterial Susceptibility to Antibiotics 

The effect of mutation on susceptibility of tested strains to different antibiotics was characterized by determining both the minimum inhibitory concentrations (MICs) and minimum biofilm inhibitory concentrations (MBICs) of tested antibiotics. These antibiotics include ampicillin, ampicillin/sulbactam, amoxicillin/clavulanic acid, piperacillin, azetronam, imipenem, cephardine, ceftazidime, cefotaxime, cefepime, ciprofloxacin, levofloxacin, gatifloxacin, tobramycin, gentamycin, tetracycline, chloramphenicol and trimethoprim/sulfamethoxazole. The broth microdilution method was employed according to Clinical Laboratory and Standards Institute Guidelines (CLSI, 2015) to determine the MICs of tested strains to different antibiotics [51,52]. MBICs are determined by broth dilution method as described earlier [47,53]. Briefly, the optical densities of overnight cultures from tested strains were adjusted equivalent to 0.5 McFarland standard. Aliquots (100 μL) of the cultures were transferred to the wells of microtiter plates and incubated overnight at 37 °C. The plates were washed with PBS and dried, and serial dilutions of antibiotics in MH broth were added to wells containing adhered biofilms. After overnight incubation at 37 °C, MBICs were considered as the lowest concentrations of antibiotics that showed no visible growth in the wells. Both positive control (inoculated bacteria in broth without addition of antibiotics) and negative control (sterile broth without bacteria) were included in the experiment. The antibiotics susceptibility experiment was repeated in triplicate.

### 2.7. In Vivo Assessment of the Pathogenesis of Tested Mutants

The influence of *sdiA* mutation on *S. enterica* pathogenesis was characterized in vivo in mice by using the protective assay, as described previously [10,54,55]. Briefly, the cell densities of tested strains overnight cultures were adjusted to approximately 1 × 10^8^ CFU/mL in LB broth. Six groups of female albino mice with similar weights were included in the assay, each containing ten mice. The first and second groups were used as negative controls, where mice were intraperitoneally injected with 100 μL PBS or kept uninoculated. Mice in the third group were injected intraperitoneally with 100 µL of *S.* Typhimurium WT strain. Mice in the fourth, fifth and sixth groups were injected with 100 µL of *S.* Typhimurium ∆*ssaV*, Δ*sdiA* or ∆*ssaV*Δ*sdiA* strains, respectively. Mice in all groups were kept at room temperature, with normal feeding and aeration. Mice survival in each group was recorded daily over 5 successive days and plotted by using the Kaplan–Meier method, and significance (* *p* < 0.05) was calculated by using Log-rank test, GraphPad Prism 5.

### 2.8. Statistical Analysis

Assays were performed in triplicate, and data are presented as median and range, unless otherwise specified. The differences between *S. enterica* WT and mutant strains were analyzed by a *t*-test, using the GraphPad Prism 5 software. A two-tailed *p-*value < 0.05 was considered statistically significant.

## 3. Results

### 3.1. CRISPR/Cas9 System Targets sdiA

*S. enterica sdiA* was targeted by a CRISPR/Cas9 system. Two guide sequences were chosen carefully to be targeted in order to achieve more efficient interference with *S. enterica sdiA*. Positive clones were selected on LB containing kanamycin and chloramphenicol. In spite of large number of escapers, colony PCR using oligo I or oligo III and sdiA-Rev primers was performed (Figure 2), and negative clones were selected and further tested.

### 3.2. Functional Testing of S. enterica ∆sdiA

It has been shown that *S. enterica* SdiA detects and responds to AHL signals produced by other microbial species [56,57]. The role of SdiA in adhesion [58] and biofilm formation [59] was further characterized. To test the success of targeting *S. enterica sdiA* by CRISPR/Cas9 system, both the adhesion and biofilm-formation capabilities of *Salmonella* Δ*sdiA* were evaluated in comparison with both *S. enterica* WT and ∆*ssaV* strains. Bacterial adhesion to epithelial HeLa cells was performed in both the presence and absence of AHL (Figure 3). *S. enterica* WT, ∆*ssaV* and Δ*sdiA* strains did not exhibit adherence capability to epithelial cells in the absence of AHL. However, the adherence capacity of tested strains significantly increased in the presence of AHL (*p* < 0.0001). In the presence of AHL, the number of adhering *S. enterica* Δ*sdiA* cells was significantly lower than *S. enterica* WT and Δ*ssaV* (*p* < 0.0001). Bacterial adhesion to epithelial cells was not affected by *ssaV* mutation, and the number of adhering cells was not affected in presence of AHL (*p* = 0.085). 

Moreover, adhesion to abiotic surface and biofilm formation of *S. enterica* WT, Δ*ssaV* and Δ*sdiA* strains were tested both in the presence and absence of AHL (Figure 4). *S. enterica* WT, Δ*ssaV*, Δ*sdiA* and Δ*ssaV*Δ*sdiA* strains were cultured with or without AHL in polystyrene microtiter plate and incubated either for 1 h (for evaluation of adhesion) or for 24 h (for evaluation of biofilm formation). Importantly, AHL significantly increased the adhesion and biofilm formation of both *S. enterica* WT and Δ*ssaV*. Moreover, the adhesion and biofilm formation of *S. enterica* were significantly reduced in *S. enterica* Δ*sdiA,* as compared with WT and Δ*ssaV* strains both in the presence and absence of AHL (*p* < 0.0001). The current results demonstrate that *S. enterica* adhesion to epithelial cells was not affected by mutation in *ssaV*. Furthermore, bacterial adhesion to abiotic surface and biofilm formation were not influenced by single mutation in *ssaV*. 

### 3.3. Intercellular Survival of S. enterica ΔsdiA 

*S.**enterica* WT, Δ*ssaV*, Δ*sdiA* and Δ*ssaV*Δ*sdiA* strains were cultured in SPI1-inducing conditions, and bacterial internalization within HeLa cells or macrophage was assessed by using the gentamicin protection assay. For invasion assays, Hela cells were washed and lysed after 1 h infection with 0.1% Triton X-100 (Figure 5A). The quorum sensing mediator AHL did not increase the invasiveness of *Salmonella* strains. Interference with *sdiA* did not affect bacterial invasiveness either in the absence or presence of AHL. However, AHL did not increase invasiveness of tested strains; the invasiveness of *sdiA* mutant was significantly reduced in comparison to the WT or *ssaV* mutant strain. On the other side, *ssaV* mutation did not influence bacterial invasiveness, as compared to *S.*
*enterica* WT or *sdiA* mutant. For intracellular replication assays, bacteria-infected cells were washed and then lysed with 0.1% Triton-X-100 at 2 and 16 h post-infection (Figure 5B), and intracellular bacteria were counted. Interestingly, AHL did not enhance the invasion of *Salmonella* strains in HeLa cells or bacterial uptake by macrophage. Obviously, *ssaV* mutation significantly decreased the intercellular bacterial replication as compared to the WT and *sdiA* mutant (*p* = 0.0062 and 0.0094; respectively). Moreover, *sdiA* mutation did not increase *Salmonella* intracellular replication within raw macrophage, as compared to *Salmonella* WT (*p* = 0.44).

### 3.4. Assessment the Expression of SPI2 Effectors S. enterica Strains

To evaluate the capability of tested strains to cope the drastic conditions inside the *Salmonella* containing vacuole (SCV) and survive in order to induce efficient immunologic response, the delivery of SPI2-effector proteins from SCV to outside by live attenuated *Salmonella* mutants (Δ*ssaV* and/or Δ*sdiA*) was used as indicator. Expression cassettes that contain *sse*J promoter were constructed with genes encoding SPI2 effector. They were used to express SPI2-T3SS translocated effector proteins SseJ tagged with HA (Figure 6A). In vitro culture conditions were used to induce both the expression of SsrAB regulon and synthesis of SPI2 effector proteins. The synthesis of SPI2-effector fusion protein tagged with HA was quantified (Figure 6B). Western blotting was employed to quantify the amounts of recombinant protein, using the Odyssey detection system and DnaK as control protein. Importantly, the expression level of recombinant protein was significantly reduced in *S.*
*enterica* Δ*ssaV* and Δ*sdiA* mutants relative to WT. 

The efficiency of the SPI2-T3SS-dependent translocation in Δ*sdiA* strain was investigated herein. *Salmonella* tested strains harboring the constructed plasmid were used to infect HeLa cells or macrophages (in presence of AHL) and then were processed for immunostaining, and the fluorescence intensities of tagged protein were measured (Figure 7A,B). The translocated proteins were significantly reduced in *S.*
*enterica ∆ssaV* and/or sdiA mutants, as compared to WT. Furthermore, the SPI2 effector translocation was significantly reduced in Δ*ssaV* or Δ*ssaV*Δ*sdiA* strains when compared to Δ*sdiA* strains. There was no difference in the SPI2-effector translocation efficacy between *S.*
*enterica* Δ*ssaV* and Δ*ssaV*Δ*sdiA* (Figure 7C,D).

### 3.5. MICs and MBICs of S. enterica Mutant Strains

The influence of the mutations on *S. enterica* resistance to antibiotics was investigated herein. The MICs and MBICs of tested antibiotics were determined by the broth microdilution method, and the results are represented in Table 2. It is shown that the MICs and MBIC were markedly decreased in *S. enterica* Δ*ssaV*, Δ*sdiA* and Δ*ssaV*Δ*sdiA* mutants in comparison to WT. This indicates that the mutation in *ssaV* and/or *sdiA* genes may increase the susceptibility and decrease the resistance to tested antibiotics. 

### 3.6. Mutation in sdiA and/or ssaV Genes Decreases S. enterica Virulence In Vivo

The impact of mutation on *S. enterica* virulence was evaluated by using mice infection models. All mice in the negative control groups survived. Similarly, all mice survived in the groups injected with *S. enterica* Δ*ssaV*, Δ*sdiA* and Δ*ssaV*Δ*sdiA*. On the other side, only five mice out 10 survived in the mice group injected with *S. enterica* WT. The mice survival was observed over five days and plotted by the Kaplan–Meier method, where significance (*p* < 0.05) was assessed by using the Log-rank test (Figure 8). These findings obviously show that the *sdiaA* and/or *ssaV* mutations markedly decreased the capacity of *S. enterica* to kill mice (*p* = 0.0069).

## 4. Discussion

*S. enterica* is an intracellular bacteria of special interest which could be engineered to deliver heterologous antigens that induce efficient cellular and humoral immune responses [18]. For this purpose, the development of specific mutations in *S. enterica* chromosome is a critical requirement [1]. In this context, this study aimed to evaluate the influence of *sdiA* mutation on the virulence of both *S. enterica* WT and *ssaV* mutant. The present findings would be valuable and extend our knowledge about employing *S. enterica* as a vector for delivering antigens and stimulating immune system.

The DNA sequences’ altering possibility within the cell in a controlled fashion greatly helps understand gene function. Importantly, the CRISPR prokaryotic immunity system has led to the identification of nucleases whose sequence specificity is programmed by small RNAs [25]. The type II CRISPR-Cas system is characterized by RNA-guided Cas9 endonuclease activity. It is the simplest of all Cas systems to be used to interfere or even edit both eukaryotic and prokaryotic genomes [31,40,41]. 

In the current work, a CRISPR-Cas9 system approach was used to target *S. enterica sdiA,* achieving efficient interference with targeted genes in two different sites. The mutation induction can be facilitated by a co-selection of transformable cells and use of dual-RNA:Cas9 cleavage to induce a small induction of recombination at the target locus [25,41]. We tried to edit a *Salmonella* chromosome to be used as a carrier for vaccination (unpublished data). Lambda red-mediated gene replacement was used to induce specific mutations; however, it was difficult to select proper tetracycline sensitive clones. In comparison, the CRISPR-Cas9 system has the advantage of being more efficient and easier as a bacterial chromosome targeting tool. These results are comparable with those reported in other studies [27,36,37,41,60].

In order to evaluate the role of SdiA in *S. enterica* pathogenesis at different stages of infection, the *sdiA* gene was targeted as described in Materials and Methods. *S. enterica* adhesion to epithelial cells and abiotic surfaces was greatly enhanced in the presence of AHL. Bacterial adhesion is the first step in biofilm formation; as AHL increases bacterial adhesion, the bacterial biofilm formation increases significantly. In order to assess the influence of AHL/SdiA on early stages of *S. enterica* invasion, the experimental conditions were adjusted to induce SPI1 effectors. As previously mentioned, SdiA is a sensor to AHL; therefore, any mutation or interference within *sdiA* would impact bacterial QS. *S. enterica* Δ*sdiA* lacked the capability to adhere to epithelial cells or abiotic surfaces, and its biofilm formation diminished significantly. The decreased *S. enterica* biofilm formation upon *sdiA* mutation relative to WT could account for the lowered MICs and MBICs of tested antibiotics. These findings are in great compliance with several studies that investigated the significant role of SdiA in *Salmonella* adhesion [15,16,17,58]. 

Moreover, *S. enterica* Δ*sdiA* exhibited a significant decreased invasion capacity within HeLa cells, regardless the presence or absence of AHL. The present results meet those of an independent work in which the increased bacterial invasion was found to be *sdiA*-dependent [61]. In addition, SdiA is known to regulate seven genes in *S. enterica* upon the activation of the SdiA transcription factor by AHL. These genes are located in two different loci: the *rck* locus and the *srgE* locus [17]. The *rck* operon includes *srgA,* which encodes a disulfide bond oxidoreductase, while SrgA plays a role in folding of fimbrial subunit (PefA) that could affect adhesion [62]. The present findings clearly indicate a role of AHL-SdiA (inducer–receptor) not only in *S. enterica* adhesion but also in biofilm formation. It is worth mentioning that AHL presence did not enhance *S. enterica* WT pathogenesis; both bacterial invasion within HeLa cells and intracellular replication in raw macrophage did not increase. However, *S. enterica* invasion was shown to be influenced by *sdiA* mutation, which may lead us to ask if SdiA can be involved directly or indirectly in the SPI1-TTSS functions. Interestingly, current data show that mutation in *sdiA* did not affect *S. enterica* intracellular replication.

Furthermore, the effect of *sdiA* mutation on the functionality of SPI2-TTSS translocation system was explored. The translocation of HA-tagged SPI2-fusion protein in both *S. enterica* WT and Δ*sdiA* mutant was investigated herein. Cells were infected with an equal number of bacteria, and translocated proteins were quantified. Surprisingly, mutation in *sdiA* significantly influenced SPI2 effector translocation. That is in compliance with the fact that *srgA* in *rck* operon, which is regulated by SdiA, plays a role in folding of outer membrane component of SPI2-TTSS [63]. Moreover, the deficient adhesion and invasion may diminish the internalized bacterial cells, and, as a consequence, the expression and translocation may be reduced. SsaV is a vital component for SPI2-TT3SS machinery and essential for secretion of a lot of TTSS effectors [7]. As predicted for *S. enterica* Δ*ssaV*, adherence and invasion within epithelial cells were not affected. In contrast, bacterial intracellular replication and translocation of SPI2 effectors were significantly reduced; these findings are in agreement with previous results [32,43]. For more convenience, the virulence characteristics of *S. enterica* Δ*sdiA*Δ*ssaV* were evaluated. The adhesion, invasion and intercellular replication of the double mutant Δ*sdiA*Δ*ssaV* were greatly diminished, regardless of the presence or absence of AHL. Importantly, double mutation in *sdiA* and *ssaV* genes confers a significate protection to the infected mice. 

Attenuated *S. enterica* has been used as a carrier for heterologous antigens, activating both humoral and cellular responses. Previous studies showed that SPI2-T3SS-deficient *S. enterica* was weakened enough and could provide protection from further challenges with WT and induces the production of both secretory IgA and serum IgG against somatic O-antigen in C57BL/6mice [19]. Moreover, *S. enterica* mutants in *ssaV* or any of SPI1-TTSS genes has been efficiently used in preparation of vaccines against typhoid fever [64] or to induce chemokines [65]. On the other hand, *S. enterica* Δ*ssaV* was found to be virulent in immunocompromised C57BL/6 mice [66]. In this study, we showed the ability of tested mutants, especially *S. enterica* Δ*sdiA*Δ*ssaV,* to confer a significant mitigation of *S. enterica* pathogenesis, in comparison with WT or *ssaV* mutant strains. Thus, we need further investigations to evaluate the possibility of using this mutant as a vaccine itself or as a suitable attenuated carrier for heterologous antigens. Targeting bacterial virulence may ease the eradication of virulent bacteria by the host’s immune system [5,67].

## 5. Conclusions

In the current study, a CRISPR-Cas9 system was employed to target bacterial chromosome efficiently. Investigating the virulence characteristics of *S. enterica* Δ*sdiA*Δ*ssaV* demonstrates that *ssaV* mutation did not influence either adherence or invasion of the *S. enterica* Δ*sdiA* strain. Similarly, *sdiA* mutation did not affect the intracellular behavior of Δ*ssaV* strain. These findings could suggest that these two virulence machineries work apart from each other, indicating that *S. enterica* Δs*diA*Δ*ssaV* requires more in vivo examination to evaluate its capability to be used as vaccine or carrier for vaccination. SdiA plays a key role in QS as a sensor to signaling AHL. Mutation of *sdiA* significantly affects bacterial adhesion and invasion, as well as biofilm formation. Current data clearly suggest that any agent that reduces SdiA transcription could be used as an antibiofilm agent that could help us control *S. enterica* infection.

## Figures and Tables

**Figure 1 microorganisms-09-02564-f001:**
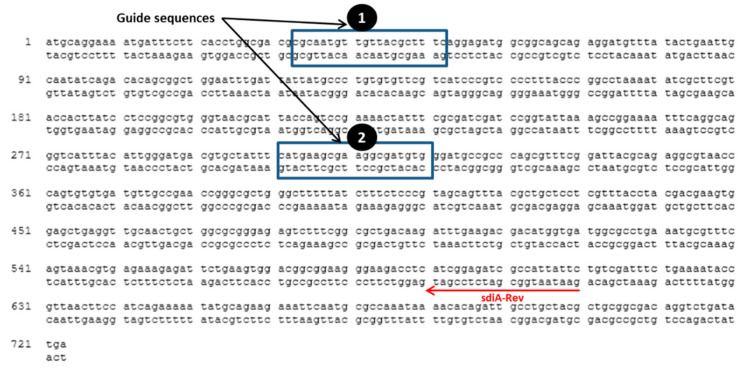
Guide sequences in *S. enterica* serovar Typhimurium NCTC 12023 *sdiA* gene (Gene ID, 1253471; NCBI Reference Sequence, NC_003197.2 (2039655..2040395). Oligo I and II were designed and annealed to target the first guide sequence; oligo III and IV were designed and annealed to target the second sequence site. For confirmation of the mutation in the selected sites, colony PCR was performed by using *sdiA*-Rev and oligo I and III as reverse and forward primers to confirm mutation in first site and second site, respectively.

**Figure 2 microorganisms-09-02564-f002:**
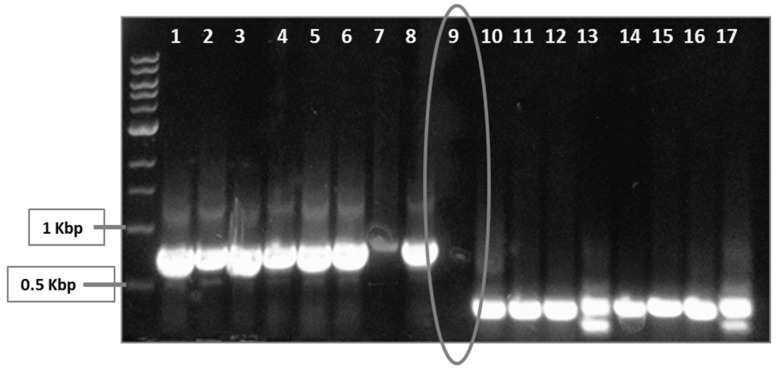
Screening for positive and negative clones, using PCR. Transformed *S. enterica* cells were selected on LB-kanamycin/chloramphenicol plates. Two plasmids were employed herein: The first plasmid, pCas9, encodes the Cas9, trcrRNA and crRNA to target guide sequence number 1. The other plasmid, pCRISPR, encodes crRNA for guide sequence number 2 to be targeted by Cas9. Both plasmids pCas9 and pCRISPR were transformed followed by selection on kanamycin and chloramphenicol-containing LB. However, there were background cells that lack the desired mutation. Colony PCR was performed by targeting both guide sequence 1, using oligo I and *sdiA*-Rev (wells 1–8), and guide sequence 2, using oligo III and *sdiA*-Rev (wells 10–17). Positive PCR clones were omitted, while negative ones (encircled well no. 9) were selected.

**Figure 3 microorganisms-09-02564-f003:**
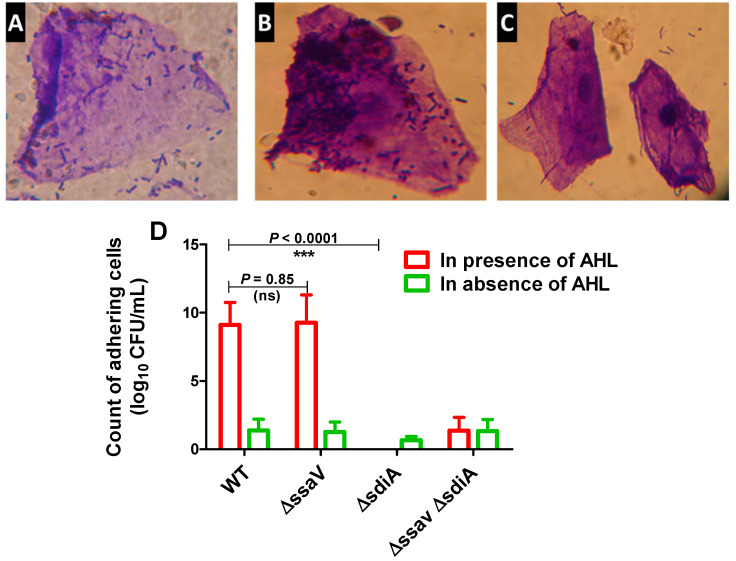
Adhesion of *S. enterica* strains to HeLa cells. HeLa cells were co-cultured with *S. enterica* WT, Δ*ssaV*, Δ*sdiA* or Δ*sdiA* Δ*ssaV* both in the presence and absence of AHL for 1 h at 37 °C. Microscopic examination of crystal violet-stained adhering *S. enterica* to HeLa cells either in presence or absence of AHL. (**A**) *S. enterica* WT adhesion in absence of AHL, (**B**) *Salmonella enterica* WT adhesion in presence of AHL and (**C**) *S. enterica* Δ*sdiA* mutant adhesion in presence of AHL. (**D**) AHL significantly increases adhesion of *S. enterica* WT and Δ*ssaV* but not Δ*sdiA* and Δ*sdiA* Δ*ssaV* mutants to Hela cells. Epithelial cells were lysed with Triton X-100 (1%). Adhering bacteria were serially diluted and counted on agar plates. Experiment was performed in triplicate, and results are represented as means ± standard deviations; *p*-value < 0.05 was considered statistically significant, using a Student’s *t*-test (*** = *p* < 0.001).

**Figure 4 microorganisms-09-02564-f004:**
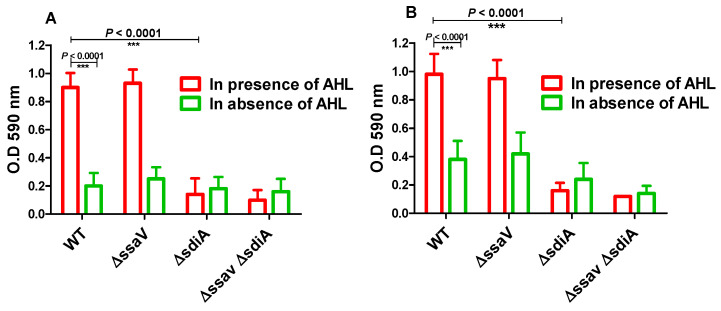
Bacterial adhesion to abiotic surface and biofilm formation. *S. enterica* WT, Δ*ssaV*, Δ*sdiA* and Δ*ssaV*Δ*sdiA* were cultured in presence or absence of AHL in polystyrene microtiter plate and incubated at 37 °C, either for 1 h (for evaluation of adhesion) or for 24 h (for evaluation of biofilm formation). (**A**) Adhering cells or (**B**) Biofilm forming cells were stained by crystal violet, ethanol was added and optical density was measured at 590 nm. Assays were performed in triplicate, and results were expressed as means ± standard deviations; *p*-value < 0.05 was considered statistically significant, using a two-tailed t-test (*** = *p* < 0.001).

**Figure 5 microorganisms-09-02564-f005:**
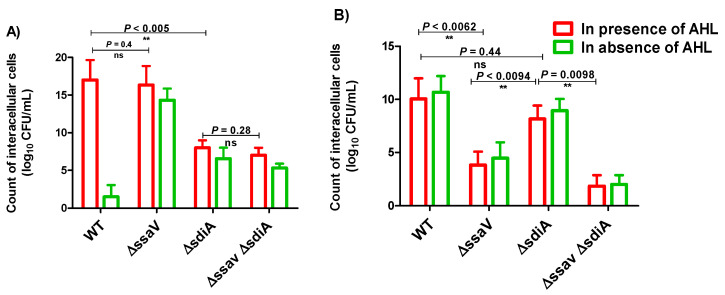
Intercellular survival of *S. enterica* strains in HeLa cells and raw macrophages. *S. enterica* WT, Δ*ssaV*, Δ*sdiA* and Δ*ssaV*Δ*sdiA* strains were cultured in suitable conditions to induce SPI1 genes. Bacterial strains were used to infect HeLa cells or raw macrophages in multiplicity of infection (MOI of 1). Non-internalized bacteria were removed by washing with PBS, and remaining extracellular bacteria were killed by using gentamicin (100 µg/mL). (**A**) Invasion assays; Hela cells were lysed with 0.1% Triton X-100 after 1 h infection, and intracellular bacteria were counted. The number of invading bacteria (1 h versus inoculum) was calculated. (**B**) Intracellular replication assays; infected cells were lysed with 0.1% Triton-X-100, and intracellular bacteria were counted at 2 and 16 h post-infection. Assays were performed in triplicate, and results were expressed as means ± standard deviations; *p*-value < 0.05 was considered statistically significant, using a two-tailed *t*-test (**= *p* < 0.01).

**Figure 6 microorganisms-09-02564-f006:**
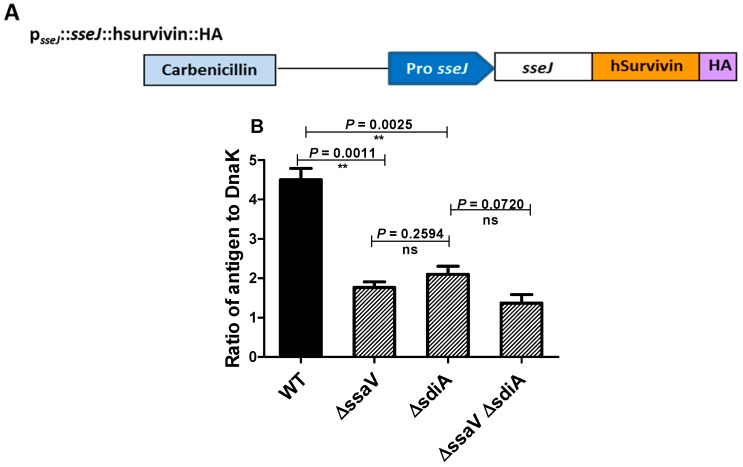
Expression of SPI2 effectors. (**A**) Plasmid-encoding SPI2 effector tagged with HA was constructed in order to test the efficacy of SPI2-TTSS-dependent translocation. (**B**) Expression of translocated proteins was evaluated as ratios of the HA to DnaK signals (**= *p* < 0.01).

**Figure 7 microorganisms-09-02564-f007:**
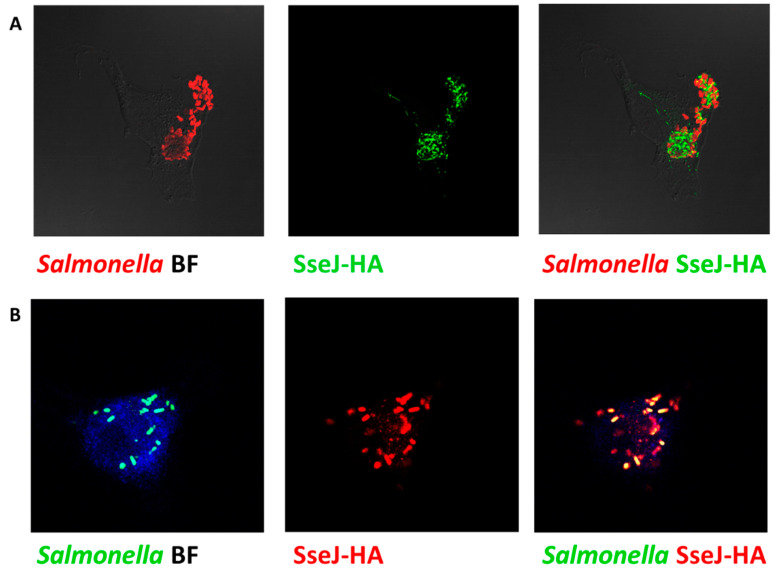
Assessment of SPI2-TTSS translocation of effector proteins. Translocation of SPI2 effectors into the cytoplasm of infected HeLa cells or raw macrophage cells (in presence of AHL) with equal number of *S. enterica* WT, Δ*ssaV*, Δ*sdiA* or Δ*ssaV*Δ*sdiA* harboring constructed plasmid was evaluated. *Salmonella* Lipopolysaccharide (rabbit anti-Salmonella O antigen) and HA epitope tag were immuno-stained and analyzed by Leica laser-scanning confocal microscope both in (**A**) HeLa cells and(**B**) macrophages. Fluorescence intensities of tagged protein were measured by using J-image program both in (**C**) HeLa cells and (**D**) macrophages. (*** = *p* < 0.001; **= *p* < 0.01; * *p* < 0.05).

**Figure 8 microorganisms-09-02564-f008:**
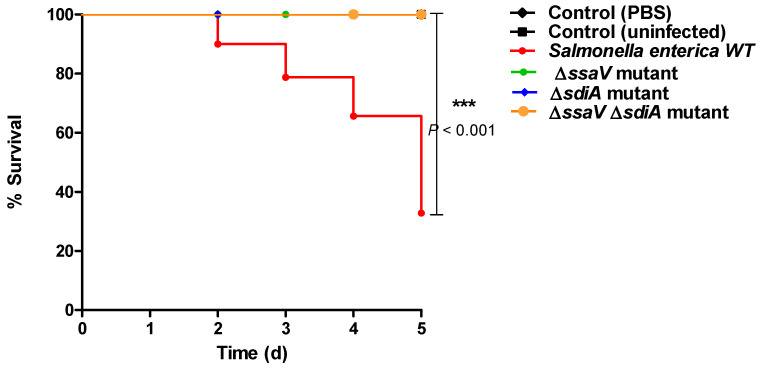
Mutation of *S. enterica sdiA* and/or *ssaV* genes significantly reduced bacterial virulence in mice. Mice (n = 10 mice/group) were injected with 100 μL of bacterial cells (2 × 10^6^ CFU/mL) of *S.* Typhimurium WT, Δ*ssaV*, Δ*sdiA* or Δ*ssaV*Δ*sdiA* strains. No death was reported for mice in negative controls, either uninfected or injected with PBS. Similarly, all mice survived in groups injected with *S. enterica* Δ*ssaV*, Δ*sdiA* and Δ*ssaV*Δ*sdiA*. In contrast, mice injected with *S. enterica* WT showed a higher mortality rate; 5 mice killed out of 10 mice (*** = *p* < 0.001).

**Table 1 microorganisms-09-02564-t001:** Oligonucleotides used in this study.

Oligonucleotide	Sequence (5′–3′)
Oligo I	AAAC CGCAATGTTGTTACGCTTTC G
Oligo II	AAAAC GAAAGCGTAACAACATTGCG
Oligo III	AAAC CATGAAGCGAAGGCGATGTG G
Oligo IV	AAAAC CACATCGCCTTCGCTTCATG
*sdiA*-Rev	GAA TAA TGG CGA TCT CCG AT
Seq-primer	CCATAAAATATGCAGGAAA
hSurv-For-EcoRV	TACGATATCGGTGCCCCGACGTTGCCCCC
hSurvivin-HA-Rev-XbaI	ATTTCTAGATTAAGCGTAGTCTGGGACGTCGTATGGGTAATCCATAGCAGCCAGCTGCTC
SseJ-Pro-For-KpnI	TACGGTACCTCACATAAAACACTAGCAC
SseJ-Rev-EcoRV	ACGGATATCTTCAGTGGAATAATGATGAGC
T7-Seq	TAATACGACTCACTATAGGG
T3-Seq	AATTAACCCTCACTAAAGG

**Table 2 microorganisms-09-02564-t002:** MICs and MBICs (µg/mL) of tested antibiotics against *S.* Typhimurium strains *.

Antibiotic	WT		Δ*ssaV*		Δ*sdiA*		Δ*ssaV*Δ*sdiA*	
	MIC	MBIC	MIC	MBIC	MIC	MBIC	MIC	MBIC
Ampicillin	256	2048	128	2048	128	1024	64	512
Ampicillin/Sulbactam	128	1024	32	512	32	512	16	128
Amoxicillin/clavulanic acid	128	1024	64	512	32	512	32	256
Piperacillin	32	256	8	32	8	16	8	16
Azetronam	32	512	32	256	16	128	8	128
Imipenem	4	8	2	4	4	4	2	4
Cephardine	64	512	16	256	32	512	16	256
Ceftazidime	32	1024	8	256	8	128	4	64
Cefotaxime	16	256	4	64	4	64	4	64
Cefepime	8	128	4	32	2	16	2	16
Ciprofloxacin	8	12	2	4	2	4	1	2
Levofloxacin	4	16	2	4	1	2	1	2
Gatifloxacin	4	16	2	8	2	8	1	4
Tobramycin	16	512	2	128	2	64	2	64
Gentamycin	16	512	2	64	2	32	2	32
Tetracycline	64	1024	8	512	16	512	8	512
Chloramphenicol	64	1024	16	512	8	256	8	256
Trimehoprim/Sulfamethoxazole	128	2048	64	1024	32	512	16	512

* Statistical analysis by Mann–Whitney U analysis demonstrates a significant difference (* *p* < 0.05) between *S.* Typhimurium WT and mutants (ΔssaV, ΔsdiA and ΔssaVΔsdiA) in their susceptibilities (MICs and MBCs) to tested antibiotics.

## Data Availability

Not applicable.

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
