# Peer review of "Alteration of Salmonella enterica Virulence and Host Pathogenesis through Targeting sdiA by Using the CRISPR-Cas9 System"

_microorganisms, 2021, doi:10.3390/microorganisms9122564_

Round 1
Reviewer 1 Report
Microorganisms Manuscript
General comments:
-In general, this manuscript has a valuable topic. The topic is scientifically sound.
-The writing style and English language needs moderate changes.
experimental design is adequate. My main concerns were introduction, data presentation and Discussion section
-There are some minor comments.
Detailed comments:
-In general, please avoid using personal pronouns such as we, our results, our work and apply this rule throughout the manuscript (for example -Line 491: our findings, Line 498: we used and line470: we investigated), and more).
Abstract:
-This section is well written.
Key words; Please add the word pathogenesis to the keywords list.
Introduction:
The topic is very important and has a great value. I see that the introduction didn’t provide enough background about the topic and needs to be enriched,
Materials and Methods:
The experimental design was suitable and adequate to the current study.
Results:
-I found that the results section is well presented but the discussion was very poor.
- Table 2: please clarify what are the data in table 2 represent (what are those values; means. Range or exact values)
-Also, statistical analysis is needed for the data in Table2.
- Figure 6: Line 337-342. The legend is too long. Please shorten it and move the details to the materials and methods section.
Discussion
I had a hard time to relate the discussion section with the corresponding data in the tables and figures, please rewrite this section and provide the appropriate citations in argument, and valuable discussion to the current results.
Conclusion:
This section is well written. The conclusion is supported by the results and includes the most significant findings.
References:
The authors provided enough citations, and it is up to Date.
***This manuscript is very valuable and will be suitable to be published in the microorganisms after minor revision.
Author Response
Dear Reviewer,
We appreciate your valuable, constructive comments and suggestions, which greatly helped us improve the manuscript. We highly appreciate your kind words which really support us. In light of your comments, we revised the manuscript.
Best Regards,
Response to Reviewer 1
We appreciate the reviewer for valuable, constructive comments and suggestions, which greatly helped us improve the manuscript.
Reviewer: This manuscript is very valuable and will be suitable to be published in the microorganisms after minor revision.
Authors: We are very thankful and grateful for the reviewer and highly appreciated your kind words which really supports us. In the light of the reviewer’s comments, we revised our manuscript as follows:
#1) Experimental design is adequate. My main concerns were introduction, data presentation and Discussion section
We appreciate the reviewer's interest in our manuscript, and we will do our best to improve the presentation and discussion of our results as suggested. All modified parts will be highlighted in the revised version of manuscript.
#2) In general, please avoid using personal pronouns such as we, our results, our work and apply this rule throughout the manuscript (for example -Line 491: our findings, Line 498: we used and line470: we investigated), and more).
We are thankful for the reviewer for drawing our attention. We will trace all these personal pronouns and correct it.
#3) Key words; Please add the word pathogenesis to the keywords list.
Added
#4) Introduction: The topic is very important and has a great value. I see that the introduction didn’t provide enough background about the topic and needs to be enriched,
We highly appreciate the reviewer's valuable comment. The introduction has been modified in the revised manuscript in order to provide more background.
#5.1) Results: I found that the results section is well presented but the discussion was very poor.
We highly appreciate the reviewer's interest in our manuscript and we improved the discussion section and highlighted changes in revised version of manuscript.
#5.2) Table 2: please clarify what are the data in table 2 represent (what are those values; means. Range or exact values)
The data in the Table 2 are the values of MIC in µg/mL and we clarified this the title of Table 2
#5.3) Also, statistical analysis is needed for the data in Table2.
The authors are very grateful for the reviewer for this valuable comment. MICs and MBCs have been determined for tested antibiotics against Salmonella strains (WT, ΔssaV, ΔsdiA, & ΔssaVΔsdiA) included in this study. MICs and MBCs reflect the lowest concentrations of antibiotics that inhibit bacterial growth. Statistical analysis Table 2 results have already performed and the authors apologize for not representing this in the first submitted version of the manuscript. Statistical analysis of table 2 results demonstrates significant differences (P < 0.05) between Salmonella WT and other mutants (ΔssaV, ΔsdiA, & ΔssaVΔsdiA) in their susceptibilities (MICs and MBCs) to most of tested antibiotics. Based on the reviewer’s comment, the statistical analysis result was added as a footnote under Table 2.
#5.4) Figure 6: Line 337-342. The legend is too long. Please shorten it and move the details to the materials and methods section.
Done.
#5.4) Discussion I had a hard time to relate the discussion section with the corresponding data in the tables and figures, please rewrite this section and provide the appropriate citations in argument, and valuable discussion to the current results.
We highly appreciate the reviewer's valuable comments. The discussion section has been modified and revised in light of the reviewer’s precious suggestions.
Finally, we really thank the reviewer for kind and valuable comments and suggestions.
Reviewer 2 Report
In the study, authors used CRISPR-Cas9 system to target the signal molecule receptor encoding gene (sdiA) in Salmonella entarica WT and ΔssaV mutant. Results showed that strains with sdiA deleted gene exabit less cell adhesion/invasion and biofilm formation but no changes for bacterial intracellular survival.
Experiments are well done and authors performed a complete investigation, however, the structure and the content of the text should be improved.
Abstract: Authors did not include the impact of sdiA mutation on Antimicrobial resistance profile. They should shorten the beginning of the paragraph and complete the abstract with all the study results. They did not also include results regarding Δssa and ΔsdiA integrations
Materials and Methods
Title 2.2 (page 3): For non-CRISPR experts readers, no Figure 1 nor Table 1, are sufficiently informative. Also, the paragraph is only enumerating oligos and sequences. Authors should describe the methodology used by steps or add a figure that could be more helpful.
Fig.7 (A and B) : Legends for each photo is unreadable
Discussion: The first half of discussion (4 paragraphs) was devoted to general information that could be simply described in the introduction. Discussion should address the interpretation of the results in the light of current knowledge. Discussion should be started at line 431 by comparing results to literature and perform a critical study of results
Author Response
Dear Reviewer,
We appreciate your valuable constructive comments and suggestions, which greatly helped us improve the manuscript. In light of your comments, we revised our manuscript.
Response to Reviewer 2
We appreciate the reviewer for valuable constructive comments and suggestions, which greatly helped us improve the manuscript. In the light of the reviewer’s comments, we revised our manuscript as follows:
#1) Experiments are well done and authors performed a complete investigation, however, the structure and the content of the text should be improved.
We are very grateful to the reviewer for valuable comment. The manuscript has been revised and changes are highlighted taking into account all comments and suggestions raised by the reviewers.
#2) Abstract: Authors did not include the impact of sdiA mutation on Antimicrobial resistance profile. They should shorten the beginning of the paragraph and complete the abstract with all the study results. They did not also include results regarding Δssa and ΔsdiA integrations
We highly appreciate the reviewer's suggestion and completely agree with him about these points. The abstract was modified in the revised version of manuscript considering all points suggested by the reviewer.
#3) Materials and Methods: Title 2.2 (page 3): For non-CRISPR experts readers, no Figure 1 nor Table 1, are sufficiently informative. Also, the paragraph is only enumerating oligos and sequences. Authors should describe the methodology used by steps or add a figure that could be more helpful.
We are thankful to the reviewer for helping us improve the manuscript. We added the needed details (yellow highlighted) in the Materials and Methods section and in the legend of Figure 1 in the revised version of manuscript as suggested.
#4) Fig.7 (A and B) : Legends for each photo is unreadable
Done, the legends were magnified in larger font.
#5) Discussion: The first half of discussion (4 paragraphs) was devoted to general information that could be simply described in the introduction. Discussion should address the interpretation of the results in the light of current knowledge. Discussion should be started at line 431 by comparing results to literature and perform a critical study of results
We appreciate the reviewer's interest in our manuscript. The discussion has been revised in order to improve the presentation of our results as suggested. All modified parts are highlighted in the revised version of manuscript.
Finally, we really thank the reviewer for kind and critical comments.
Best Regards,
Reviewer 3 Report
It is widely known that quorum sensing is a key regulator of Salmonella virulence. Several studies have determined that this receptor is key in the expression of certain Salmonella virulence and biofilm factors. It is also widely known the potential of CRISPR-Cas9 system for gene editing. In general terms, I do not see a clear objective in the work. What is the real goal of editing the sdiA gene? If the objective is to edit the genome in order to have a bacterium that can be used to deliver heterolgeous antigens, it is certainly not the right method as it reduces the invasion of Salmonella. For these reasons, from my point of view, the publication of this work is not justified.
I recommend to the authors to fully revise the material and methods section before sending a manuscript. The methods have to be perfectly reproduced by other researchers. For example, an important point as AHL concentration and how it was prepared is not provided in any point of the manuscript
Other comments:
Line 100: why one or other antibiotic? What medium was used. This section and the reagents used is not properly described. There is a lack of information.
Line 117: Growth conditions?
Table 1: Do the authors consider that Oligo I, II, III, IV is a correct name? I do not know for what these primers were used for.
Line 121: Accesion number for sdiA sequence?
Line 124: Write properly the serotype name.
Line 127: Describe the assay please.
Line 130: Serotype name. What concentration of bacteria were used, what concentration of AHLs were used. This information is basic and the authors should provided since the first time this information. It is important to revise these things before sending a manuscript for review. It is not properly described how the HeLa cells were prepared.
Line 139: The same comment about AHLs concentrations.
Line 140: plates.
Line 154: The same comment about AHLs concentrations.
Line 203: What antimicrobials were tested?
Line 209: Delete this sentence. It is a repetition of the definition provided before.
Line 216: So, what was positive control?
Figure 2: Is electrophoresis described in Material and methods section?
Figure 3: Please provide information of Y axys. Is it Salmonella count? CFUs?
Figure 5: I don’t understand Y axys….
Author Response
Dear Reviewer,
We appreciate your valuable constructive comments and suggestions, which greatly helped us improve the manuscript. In light of your comments, we revised our manuscript.
Response to Reviewer 3
We appreciate the reviewer for valuable and constructive comments and suggestions, which greatly helped us improve the manuscript. In the light of the reviewer’s comments, we revised our manuscript as follows:
#1) It is widely known that quorum sensing is a key regulator of Salmonella virulence. Several studies have determined that this receptor is key in the expression of certain Salmonella virulence and biofilm factors. It is also widely known the potential of CRISPR-Cas9 system for gene editing. In general terms, I do not see a clear objective in the work. What is the real goal of editing the sdiA gene? If the objective is to edit the genome in order to have a bacterium that can be used to deliver heterolgeous antigens, it is certainly not the right method as it reduces the invasion of Salmonella. For these reasons, from my point of view, the publication of this work is not justified.
We highly appreciate the reviewer's comment and find it is a good chance for fruitful discussion to clarify our point of view and improve the manuscript. We totally agree with reviewer’s comment that SdiA function in Salmonella is known and employing that CRISPER technique in editing bacterial genomes is known as well. However, we aimed in this study to evaluate the effect of sdiA mutation on bacterial virulence. Salmonella virulence is owed to its adhesion to the host cells followed by its invasion as intracellular parasite. For this purpose, Salmonella uses 2 distinct types of TTSS systems; one system to ease its invasion and the second one for its survival within the phagosome. In the current study, we evaluated the effect of sdiA mutation and the effect of double mutant in both sdiA and ssaV genes on Salmonella virulence taking into consideration the crucial roles of both sdiA and ssaV in bacterial adhesion and functionality of TTSS. As far as we know, there is no previous study that evaluated the impact of this double mutant on bacterial virulence, which is our aim in this study.
- From our previous experience in developing carriers for vaccination using Salmonella, it is very critical to understand a lot of factors that we reviewed earlier (Hegazy and Hensel, 2012). Regarding the reviewer’s comment that “…….., it is certainly not the right method as it reduces the invasion of Salmonella”, in order to edit the bacterial genome to be used to deliver heterolgeous antigens, it is very important to keep a balance between Salmonella attenuation and its virulence. In other words, Salmonella should be attenuated enough in order to lack its severe pathogenicity in host; however, in the same time it should exhibits a higher capability of inducing the host immune response. In this study, our main target was to examine the virulence of mutant strains, however in order to CONFIRM the efficacy of these mutants as carriers for vaccination, it is needed to evaluate their capacity to induce CD4-Tcells and CD8-T cells as we have shown previously (Hegazy, et al, 2012 and 2014). Moreover, we think that these mutants are still capable of inducing the immune response in spite of their reduced invasion capacity, which strongly agrees with the hypothesis of vaccine preparation.
- Actually, we aimed in this study to evaluate the effect of mutation on Salmonella However, if we want to use these tested strains as carriers for vaccines, we will test S. Typhimurium as a model in mice as it causes infection symptoms like that S. Typhi induces in human and it is needed to test these mutations in S. Typhi and try it. From our experience, we have found that mutations that attenuate S. Typhimurium did not lead to the same degree of attenuation in S. Typhi (Xu et al, 2014 and unpublished data). In other words, the S. Typhi should be attenuated more than S. Typhimurium to avoid its virulence. That is why we targeted these two important genes in S. Typhimurium prior to test them in S. Typhi.
Conclusively, this study has obviously revealed the exact role of sdiA in Salmonella pathogenesis and confirmed that CRISPER is a successful technique for genome editing. However, we aimed to elucidate how much the mutation in sdiA, ssaV alone or in both genes would affect Salmonella virulence. This work is a preliminarily study in which we screened numerous Salmonella virulence genes to be targets for successful mutation that keeps the balance: attenuated enough (not highly pathogenic) and virulent enough (capable of inducing the immune response). We confess that we did not present our objective in this study clearly; however, this will be done in the modified version of manuscript as advised by the reviewer.
References:
- Xu X, Hegazy WA, Guo L, Gao X, Courtney AN, Kurbanov S, Liu D, Tian G, Manuel ER, Diamond DJ, Hensel M, Metelitsa LS. Effective cancer vaccine platform based on attenuated salmonella and a type III secretion system. Cancer Res. 2014 Nov 1;74(21):6260-70. doi: 10.1158/0008-5472.CAN-14-1169. Epub 2014 Sep 11. PMID: 25213323; PMCID: PMC4216746.
- Hegazy WA, Xu X, Metelitsa L, Hensel M. Evaluation of Salmonella enterica type III secretion system effector proteins as carriers for heterologous vaccine antigens. Infect Immun. 2012 Mar;80(3):1193-202. doi: 10.1128/IAI.06056-11. Epub 2012 Jan 17. Erratum in: Infect Immun. 2015 Mar;83(3):1225. PMID: 22252866; PMCID: PMC3294654.
- Hegazy WA, Hensel M. Salmonella enterica as a vaccine carrier. Future Microbiol. 2012 Jan;7(1):111-27. doi: 10.2217/fmb.11.144. PMID: 22191450.
#2) I recommend to the authors to fully revise the material and methods section before sending a manuscript. The methods have to be perfectly reproduced by other researchers. For example, an important point as AHL concentration and how it was prepared is not provided in any point of the manuscript
We highly appreciate the reviewer's comment and very thankful for drawing our attention for such points. We completely agree, we will revise the methods section carefully and add needed details in the revised version.
The source and concentration of AHL used in this study will be added to the revised manuscript. AHLs have different affinities by SdiA, we used in this study N-hexanoyl-DL-homoserine lactone (Sigma-Aldrich, USA), CAS Number: 106983-28-2, at concentrations of 0.001 µM. There are various studies that explored the affinities of different AHLs to SdiA as reviewed by Almeida et al, 2018 and Michael et al., 2001. However, oxoC8 has been shown to be the most potent AHL that can bind to SdiA in very low concentrations ranged from 0.01-0.001 µM (Smith et al., 2008; Michael et al., 2001)
References:
- Almeida FA, Vargas ELG, Carneiro DG, Pinto UM, Vanetti MCD. Virtual screening of plant compounds and nonsteroidal anti-inflammatory drugs for inhibition of quorum sensing and biofilm formation in Salmonella. Microb Pathog. 2018 Aug;121:369-388. doi: 10.1016/j.micpath.2018.05.014. Epub 2018 May 12. PMID: 29763730.
- Smith JN, Dyszel JL, Soares JA, Ellermeier CD, Altier C, Lawhon SD, Adams LG, Konjufca V, Curtiss R 3rd, Slauch JM, Ahmer BM. SdiA, an N-acylhomoserine lactone receptor, becomes active during the transit of Salmonella enterica through the gastrointestinal tract of turtles. PLoS One. 2008 Jul 30;3(7):e2826. doi: 10.1371/journal.pone.0002826. PMID: 18665275; PMCID: PMC2475663.
- Michael B, Smith JN, Swift S, Heffron F, Ahmer BM. SdiA of Salmonella enterica is a LuxR homolog that detects mixed microbial communities. J Bacteriol. 2001 Oct;183(19):5733-42. doi: 10.1128/JB.183.19.5733-5742.2001. PMID: 11544237; PMCID: PMC95466.
#3) Line 100: why one or other antibiotic? What medium was used. This section and the reagents used is not properly described. There is a lack of information.
We are thankful to reviewer for helping us improving this manuscript. The section “Bacterial Strains, Plasmids and Enzymes", in the manuscript will be modified following the reviewer’s suggestions. The used media and AHLs and their sources will be added clearly in the revised manuscript. Briefly, the mentioned antibiotics were used to select bacterial cells harboring the electroporated plasmid. For instance, CRISPER plasmids harbors chloramphenicol or kanamycin resistance genes and will be selected from Mueller-Hinton (MH) agar plates provided with chloramphenicol or kanamycin. Similarly, the pWSK29 plasmid carries kanamycin resistance gene and bacterial cells carrying this plasmid will be selected from MH plates provided with kanamycin.
#4) Line 117: Growth conditions?
Growth conditions were added and highlighted in the revised manuscript.
#5) Table 1: Do the authors consider that Oligo I, II, III, IV is a correct name? I do not know for what these primers were used for.
In construction of CRISPER plasmids, we followed the protocol provided with purchased plasmids from Addgene (http://www.addgene.org/) and according to Jiang et al., 2013. First, we determined the nucleotide sequences in sdiA gene that to be targeted. These sequences matching our targets were obtained and annealed together. The annealed double stranded product was next ligated to CRISPER plasmids after digestion with BsaI-endonuclease enzyme. These plasmids were transformed to bacteria and bind specifically with the target sequence. Finally, Cas9 DNA endonuclease enzyme associated with the CRISPER system can cleave specifically these targets. Regarding the names of the used oligos, we simply targeted two sites, used oligo I and II to target site 1 and oligo III and IV to target site 2 which we designed specifically in this study as shown in Figure 1. The word "Oligos" was used to name the synthesized oligonucleotides which are used to target the specific sequences, and it was used in several previous publications (Jiang et al., 2013).
Reference
Jiang W, Bikard D, Cox D, Zhang F, Marraffini LA. RNA-guided editing of bacterial genomes using CRISPR-Cas systems. Nat Biotechnol. 2013 Mar;31(3):233-9. doi: 10.1038/nbt.2508. Epub 2013 Jan 29. PMID: 23360965; PMCID: PMC3748948.
#6) Line 121: Accesion number for sdiA sequence?
The accession number is added.
#7) Line 124: Write properly the serotype name.
We are thankful to reviewer for helping us improving this manuscript. The serotype name is corrected.
#8) Line 127: Describe the assay please.
The assay was described in the revised manuscript as advised by the reviewer in sections 2.3.1 and 2.3.2.
#9) Line 130: Serotype name. What concentration of bacteria were used, what concentration of AHLs were used. This information is basic and the authors should provided since the first time this information. It is important to revise these things before sending a manuscript for review. It is not properly described how the HeLa cells were prepared.
We are very grateful to the reviewer for helping us improving this manuscript. All points addressed by the reviewer were carefully considered in the revised manuscript as follows:
- Serotype name was corrected
- Cell density of 1 × 106 CFU/mL (OD600 = 0.4) for adhesion assay" was mentioned in the revised manuscript.
- Concentration of AHL was 0.001 µM and was clearly mentioned the revised manuscript.
- Preparation of HeLa cells: HeLa cells were passaged with 70% confluent
#10) Line 139: The same comment about AHLs concentrations.
Done.
#11) Line 140: plates.
Corrected
#12) Line 154: The same comment about AHLs concentrations.
Done.
#13) Line 203: What antimicrobials were tested?
The used antibiotics are already listed in Table 2, and we mentioned this in this section.
#14) Line 209: Delete this sentence. It is a repetition of the definition provided before.
We appreciate the reviewer's comment. This sentence was deleted as recommended.
#15) Line 216: So, what was positive control?
Positive control is the inoculated bacteria in broth without addition of antibiotics, which will grow and show visible turbidity. On the other hand, negative control is clear sterile broth without inoculation of bacteria, which remained clear showing no signs of bacterial growth. Both positive and negative controls are routinely used these experiments to compare any visually observed bacterial growth and confirm working under aseptic conditions. And as recommended by CLSI, E.coli (ATCC 25922) was used as quality control standard.
#16) Figure 2: Is electrophoresis described in Material and methods section?
We are thankful for the reviewer's interest in our manuscript. The description of the electrophoresis was added in the revised manuscript as recommended by the reviewers. PCR products were visualized by electrophoresis on agarose gel (0.7%) using 1X TAE (Tris-acetate-EDTA) running buffer at 80-120V and visualized by 0.5 g/ml ethidium bromide.
#17) Figure 3: Please provide information of Y axys. Is it Salmonella count? CFUs?
We are very grateful to the reviewer for helping us improve the manuscript. The Y axis in Figure 3 was adjusted and it is clearly mentioned that Y axis refers to the number of adhering bacteria in CFU/mL.
#18) Figure 5: I don’t understand Y axys….
The Y axis in figure 5 represents the numbers of intracellular Salmonella bacteria. The numbers of intracellular Salmonella of tested strains in HeLa cells and macrophages were determined as described before (Hölzer and Hensel, 2012). Briefly, following bacterial adhesion, the extracellular bacteria were removed and HeLa cells and macrophages were lysed. The intracellular bacteria were serially diluted, plated on agar plates. The viable cells were counted and represented as CFU/mL.
Reference:
Hölzer SU, Hensel M. Divergent roles of Salmonella pathogenicity island 2 and metabolic traits during interaction of S. enterica serovar typhimurium with host cells. PLoS One. 2012;7(3):e33220. doi: 10.1371/journal.pone.0033220. Epub 2012 Mar 12. PMID: 22427996; PMCID: PMC3299762.
We hope the reviewer could accept our explanation
Finally, we really thank the reviewer for kind and critical comments.
Best Regards,
Round 2
Reviewer 3 Report
The authors have significantly improved the manuscript following the comments made by the reviewers and have changed my opinion about the article